PBxplore: a tool to analyze local protein structure and deformability with Protein Blocks

Barnoud Jonathan 1 2 3 4 6
Santuz Hubert 1 2 3 4 7
Craveur Pierrick 1 2 3 4 8
Joseph Agnel Praveen 1 2 3 4 9
Jallu Vincent 5
de Brevern Alexandre G. alexandre.debrevern@univ-paris-diderot.fr 1 2 3 4
Poulain Pierre pierre.poulain@univ-paris-diderot.fr 1 2 3 4 10
1 INSERM, U 1134, DSIMB , Paris , France
2 Univ. Paris Diderot, Sorbonne Paris Cité, Univ de la Réunion, Univ des Antilles, UMR-S 1134 , Paris , France
3 Institut National de la Transfusion Sanguine (INTS) , Paris , France
4 Laboratoire d’Excellence GR-Ex , Paris , France
5 Platelet Unit, INTS , Paris , France
6 Current affiliation:  Groningen Biomolecular Sciences and Biotechnology Institute and Zernike Institute for Advanced Materials, University of Groningen , Groningen , The Netherlands
7 Current affiliation:  Laboratoire de Biochimie Théorique, CNRS UPR 9080, Institut de Biologie Physico-Chimique , Paris , France
8 Current affiliation:  Department of Integrative Structural and Computational Biology, The Scripps Research Institute , La Jolla , CA , United States of America
9 Current affiliation:  Birkbeck College, University of London , London , UK
10 Current affiliation:  Mitochondria, Metals and Oxidative Stress Group, Institut Jacques Monod, UMR 7592, Univ. Paris Diderot, CNRS, Sorbonne Paris Cité , Paris , France
de Azevedo Jr Walter
Electronic publication date: 2017 Nov 20
Publication date: 2017
Volume: 5
Electronic Location ID: e4013
Received 2017 Aug 28; Accepted 2017 Oct 19
Copyright: ©2017 Barnoud et al.
Copyright year: 2017
Copyright holder: Barnoud et al.
License: This is an open access article distributed under the terms of the Creative Commons Attribution License, which permits unrestricted use, distribution, reproduction and adaptation in any medium and for any purpose provided that it is properly attributed. For attribution, the original author(s), title, publication source (PeerJ) and either DOI or URL of the article must be cited.
License URL: https://creativecommons.org/licenses/by/4.0/

Keywords: Protein blocks, Deformability, Python, Protein, Structure, Structural alphabet

Funding: National Institute for Blood Transfusion Lab of Excellence GR-Ex ANR-11-LABX-0051 Ministry of Research University Paris Diderot Sorbonne Paris Cité National Institute for Health and Medical Research French National Research Agency ANR-11-IDEX-0005-02 Indo-French Centre for the Promotion of Advanced Research/CEFIPRA 5302-2 Netherlands Organisation for Scientific Research (NWO) This work was supported by grants from the National Institute for Blood Transfusion (INTS, France) and the Lab of Excellence GR-Ex to Jonathan Barnoud, Hubert Santuz, Pierrick Craveur, Agnel Praveen Joseph, Vincent Jallu, Alexandre G. de Brevern and Pierre Poulain; and from the Ministry of Research (France), University Paris Diderot, Sorbonne Paris Cité (France), National Institute for Health and Medical Research (INSERM, France) to Jonathan Barnoud, Hubert Santuz, Pierrick Craveur, Agnel Praveen Joseph, Alexandre G. de Brevern and Pierre Poulain. The labex GR-Ex, reference ANR-11-LABX-0051 is funded by the program “Investissements d’avenir” of the French National Research Agency, reference ANR-11-IDEX-0005-02. Alexandre G. de Brevern was also supported by the Indo-French Centre for the Promotion of Advanced Research/CEFIPRA grant (number 5302-2). Jonathan Barnoud was also supported by the TOP program of Prof. Marrink, financed by the Netherlands Organisation for Scientific Research (NWO). The funders had no role in study design, data collection and analysis, decision to publish, or preparation of the manuscript.

==============================
This paper describes the development and application of a suite of tools, called PBxplore, to analyze the dynamics and deformability of protein structures using Protein Blocks (PBs). Proteins are highly dynamic macromolecules, and a classical way to analyze their inherent flexibility is to perform molecular dynamics simulations. The advantage of using small structural prototypes such as PBs is to give a good approximation of the local structure of the protein backbone. More importantly, by reducing the conformational complexity of protein structures, PBs allow analysis of local protein deformability which cannot be done with other methods and had been used efficiently in different applications. PBxplore is able to process large amounts of data such as those produced by molecular dynamics simulations. It produces frequencies, entropy and information logo outputs as text and graphics. PBxplore is available at https://github.com/pierrepo/PBxplore and is released under the open-source MIT license.

Introduction

Proteins are highly dynamic macromolecules (Frauenfelder, Sligar & Wolynes, 1991; Bu & Callaway, 2011). To analyze their inherent flexibility, computational biologists often use molecular dynamics (MD) simulations. The quantification of protein flexibility is based on various methods such as Root Mean Square Fluctuations (RMSF) that rely on multiple MD snapshots or Normal Mode Analysis (NMA) that rely on a single structure and focus on quantifying large movements.

Alternative in silico approaches assess protein motions through the protein residue network (Atilgan, Turgut & Atilgan, 2007) or dynamical correlations from MD simulations (Ghosh & Vishveshwara, 2007; Dixit & Verkhivker, 2011). Another noticeable development is the MOdular NETwork Analysis (MONETA), which localizes the perturbations propagation throughout a protein structure (Laine, Auclair & Tchertanov, 2012).

Here we use an alternative yet powerful approach based on small prototypes or “structural alphabets” (SAs). SAs approximate conformations of protein backbones and code the local structures of proteins as one-dimensional sequences (Offmann, Tyagi & De Brevern, 2007). Protein Blocks (PBs) (De Brevern, Etchebest & Hazout, 2000) are one of these SAs (De Brevern, 2005; Etchebest et al., 2005; Joseph et al., 2010).

Figure 1 (A) The 16 protein blocks (PBs) represented in balls with carbon atoms in gray, oxygen atoms in red and nitrogen atoms in purple (hydrogen atoms are not represented). (B) The barstar protein (PDB ID 1AY7 (Sevcík et al., 1998)) represented in cartoon with alpha-helices in blue, beta-strands in red and coil in pink. These representations were generated using PyMOL software (DeLano, 2002) (C) PBs sequence obtained from PBs assignment. Z is a dummy PB, meaning that no PB can be assigned to this position.

PBs are composed of 16 blocks designed through an unsupervised training performed on a representative non-redundant databank of protein structures (De Brevern, Etchebest & Hazout, 2000). PBs are defined from a set of dihedral angles describing the protein backbone. This property makes PBs interesting conformational prototypes of the local protein structure. PBs are labeled from a to p (see Fig. 1A). PBs m and d are prototypes for central α-helix and central β-strand, respectively. PBs a to c primarily represent β-strand N-caps and PBs e and f, β-strand C-caps; PBs g to j are specific to coils, PBs k and l are specific to α-helix N-caps, and PBs n to p to α-helix C-caps (De Brevern, 2005). Figure 1 illustrates how a PB sequence is assigned from a protein structure. Starting from the 3D coordinates of the barstar protein (Fig. 1B), the local structure of each amino acid is compared to the 16 PB definitions (Fig. 1A). The most similar protein block is assigned to the residue under consideration (the similarity metric is explained latter in this article). Eventually, assignment leads to the PB sequence represented in Fig. 1C.

By reducing the complexity of protein structure, PBs have been shown to be efficient and relevant in a wide spectrum of applications. To name a few, PBs have been used to analyze protein contacts (Faure, Bornot & De Brevern, 2008), to propose a structural model of a transmembrane protein (De Brevern, 2005), to reconstruct globular protein structures (Dong, Wang & Lin, 2007), to design peptides (Thomas et al., 2006), to define binding site signatures (Dudev & Lim, 2007), to perform local protein conformation predictions (Li, Zhou & Liu, 2009; Rangwala, Kauffman & Karypis, 2009; Suresh, Ganesan & Parthasarathy, 2013; Suresh & Parthasarathy, 2014; Zimmermann & Hansmann, 2008), to predict β-turns (Nguyen et al., 2014) and to understand local conformational changes due to mutations of the αIIb β3 human integrin (Jallu et al., 2012; Jallu et al., 2013; Jallu et al., 2014).

PBs are also useful to compare and superimpose protein structures with pairwise and multiple approaches (Joseph, Srinivasan & De Brevern, 2011; Joseph, Srinivasan & De Brevern, 2012), namely iPBA (Gelly et al., 2011) and mulPBA (Léonard et al., 2014), both currently showing the best results compared to other superimposition methods. Eventually, PBs lead to interesting results at predicting protein structures from their sequences (Ghouzam et al., 2015; Ghouzam et al., 2016) and at predicting protein flexibility (Bornot, Etchebest & de Brevern, 2011; De Brevern et al., 2012).

Applying PB-based approaches to biological systems such as the DARC protein (De Brevern et al., 2005), the human αIIb β3 integrin (Jallu et al., 2012; Jallu et al., 2013; Jallu et al., 2014) and the KISSR1 protein (Chevrier et al., 2013) highlighted the usefulness of PBs in understanding local deformations of large protein structures. Specifically, these analyzes have shown that a region considered as highly flexible through RMSF quantifications can be seen using PBs as locally highly rigid. This unexpected behavior is explained by a local rigidity surrounded by deformable regions (Craveur et al., 2015). To go further, we recently used PBs to analyze long-range allosteric interactions in the Calf-1 domain of αIIb integrin (Goguet et al., 2017). To our knowledge, the only other related approach based on SA to assess local deformation is GSATools (Pandini et al., 2013); it is specialized in the analysis of functional correlations between local and global motions, and the mechanisms of allosteric communication.

Despite the versatility of PBs and the large spectrum of their applications, PBs lack a uniform and easy-to-use toolkit to assign PB sequences from 3D structures, and to analyze these sequences. The only known implementation is a an old C program not publicly available and not maintained anymore. Such a tool not being available limits the usability of the PBs for studies where they would be meaningful.

We thus propose PBxplore, a tool to analyze local protein structure and deformability using PBs. It is available at https://github.com/pierrepo/PBxplore. PBxplore can read PDB structure files (Bernstein et al., 1977), PDBx/mmCIF structure files (Bourne et al., 1997), and MD trajectory formats from most MD engines, including Gromacs MD topology and trajectory files (Lindahl, Hess & Van der Spoel, 2001; Van der Spoel et al., 2005). Starting from 3D protein structures, PBxplore assigns PBs sequences; it computes a local measurement of entropy, a density map of PBs along the protein sequence and a WebLogo-like representation of PBs.

In this paper, we first present the principle of PBxplore, then its different tools, and finally a step-by-step user-case with the β3 subunit of the human platelet integrin αIIb β3.

Design and Implementation

PBxplore is written in Python (Van Rossum, 1995; Python Software Foundation, 2010; Bassi, 2007). It is compatible with Python 2.7, and with Python 3.4 or greater. It requires the Numpy Python library for array manipulation (Ascher et al., 1999), the matplotlib library for graphical representations, and the MDAnalysis library for molecular dynamics simulation files input (Michaud-Agrawal et al., 2011; Gowers et al., 2016). Optionally, PBxplore functionalities can be enhanced by the installation and the use of WebLogo (Crooks et al., 2004) to create sequence logos.

PBxplore is available as a set of command-line tools and as a Python module. The command-line tools allow easy integration of PBxplore in existing analysis pipelines. These programs can be linked up together to carry out the most common analyses on PB sequences to provide insights on protein flexibility. In addition, the PBxplore Python library provides an API to access its core functionalities which allows the integration of PBxplore in Python programs and workflows, and the extension of the method to suit new needs.

PBxplore is released under the open-source MIT license (https://opensource.org/licenses/MIT). It is available on the software development platform GitHub at https://github.com/pierrepo/PBxplore.

The package contains unit and regression tests and is continuously tested using Travis CI (https:// travis-ci.org/). An extensive documentation is available on Read the Docs (Holscher, Leifer & Grace, 2010) at https://pbxplore.readthedocs.io.

Installation

The easiest way to install PBxplore is through the Python Package Index (PyPI):

pip install --user pbxplore

It will ensure all required dependencies are installed correctly.

Command-line tools

A schematic description of PBxplore command line interface is provided in Fig. 2. The interface is composed of three different programs: PBassign to assign PBs, PBcount to compute PBs frequency on multiple conformations, and PBstat to perform statistical analyses and visualization. These programs can be linked up together to make a structure analysis pipeline to study protein flexibility.

Figure 2 PBxplore is based on 3 programs that can be chained to build a structure analysis pipeline.

Main input file types (.pdb, MD trajectory, MD topology), output files (.fasta, .png, .Neq, .pdf) and parameters (beginning with a single or double dash) are indicated.

PBassign

The very first task is to assign PBs from the protein structure(s). A PB is associated to each pentapeptide in the protein chain. To assign a PB to the residue n, five residues are required (residues n − 2, n − 1, n, n + 1 and n + 2). From the backbone conformation of these five residues, eight dihedral angles (ψ and ϕ) are computed, going from the ψ angle of residue n − 2 to the ϕ angle of residue n + 2 (De Brevern, 2005). This set of eight dihedral angles is then compared to the reference angles set for the 16 PBs (De Brevern, Etchebest & Hazout, 2000) using the Root Mean Square Deviation Angle (RMSDA) measure, i.e., an Euclidean distance on angles. PB with the smallest RMSDA is assigned to residue n. A dummy PB Z is assigned to residues for which all eight angles cannot be computed. Hence, the first two N-terminal and the last two C-terminal residues are always assigned to PB Z.

The program PBassign reads one or several protein 3D structures and performs PBs assignment as one PBs sequence per input structure. PBassign can process multiple structures at once, either provided as individual structure files or as a directory containing many structure files or as topology and trajectory files obtained from MD simulations. Note that PBxplore is able to read any trajectory file format handled by the MDAnalysis library, yet our tests focused on Gromacs trajectories. Output PBs sequences are bundled in a single file in FASTA format.

PBcount

During the course of a MD simulation, the local protein conformations can change. It is then interesting to analyze them through PB description. Indeed, as each PB describes a local conformation, the variability of the PB assigned to a given residue throughout the trajectory indicates some local deformation of the protein structure. Thus, once PBs are assigned, PBs frequencies per residue can be computed.

The program PBcount reads PBs sequences for different conformations of the same protein from a file in FASTA format (as outputted by PBassign). Many input files can be provided at once. The output data is a 2D matrix of x rows by y columns, where x is the length of the protein sequence and y is the 16 distinct PBs. A matrix element is the count of a given PB at a given position in the protein sequence.

PBstat

The number of possible conformational states covered by PBs is higher than the classical secondary structure description (16 states instead of 3). As a consequence, the amount of information produced by PBcount can be complex to handle. Hence, we propose three simple ways to visualize the variation of PBs which occur during a MD simulation.

The program PBstat reads PBs frequencies as computed by PBcount. It can produce three types of outputs based on the input argument(s). The first two use the matplotlib library and the last one requires the installation of the third-party tool Weblogo (Crooks et al., 2004). PBstat also offers two options (–residue-min and –residue-max) to define a residue frame allowing the user to quickly look at segments of interest. The three graphical representations proposed are:

• Distribution of PBs. This feature plots the frequency of each PB along the protein sequence. The output file could be in format .png, .jpg or .pdf. A dedicated colorblind safe color range (Brewer et al., 2013) allows visualizing the distribution of PBs. For a given position in the protein sequence, blue corresponds to a null frequency when the particular PB is never sampled at this position and red corresponds to a frequency of 1 when the particular PB is always found at this position. This representation is produced with the –map argument.

• Equivalent number of PBs (Neq). The Neq is a statistical measurement similar to entropy (Offmann, Tyagi & De Brevern, 2007). It represents the average number of PBs sampled by a given residue. Neq is calculated as follows: Neq= exp−∑i=116fx lnfx

where fx is the probability (or frequency) of the PB x. A Neq value of 1 indicates that only a single type of PB is observed, while a value of 16 is equivalent to a random distribution, i.e., all PBs are observed with the same frequency 1/16. For example, a Neq value around 5 means that, across all the PBs observed at the position of interest, 5 different PBs are mainly observed. If the Neq exactly equals to 5, this means that 5 different PBs are observed in equal proportions (i.e., 1/5).

A high Neq value can be associated with a local deformability of the structure whereas a Neq value close to 1 means a rigid structure. In the context of structures issued from MD simulations, the concept of deformability / rigidity is independent to the one of mobility. The Neq representation is produced with the –neq argument.

• Logo representation of PBs frequency. This is a WebLogo-like representation (Crooks et al., 2004) of PBs sequences. The size of each PB is proportional to its frequency at a given position in the sequence. This type of representation is useful to pinpoint PBs patterns. This WebLogo-like representation is produced with the –logo argument.

Python module

PBxplore is also a Python module that more advanced users can embed in their own Python script. Here is a Python 3 example that assigns PBs from the structure of the barstar ribonuclease inhibitor (Lubienski et al., 1994):

import urllib.request import pbxplore as pbx # Download the pdb file urllib.request.urlretrieve (’https://files.rcsb.org/view/1BTA.pdb’, ’1BTA.pdb’) # The function pbx.chain_from_files () reads a list of files # and for each one returns the chain and its name. for chain_name, chain in pbx.chains_from_files ([’1BTA.pdb’]):     # Compute phi and psi angles     dihedrals = chain.get_phi_psi_angles ()     # Assign PBss     pb_seq = pbx.assign(dihedrals)     print (’PBs sequence for chain {}:\n {}’.format (chain_name,     pb_seq))

The documentation contains complete and executable Jupyter notebooks explaining how to use the module. It goes from the PBs assignments to the visualization of the protein deformability using the analysis functions. This allows the user to quickly understand the architecture of the module.

Results

This section aims at giving the reader a quick tour of PBxplore features on a real-life example. We will focus on the β3 subunit of the human platelet integrin αIIbβ3 that plays a central role in hemostasis and thrombosis. The β3 subunit has also been reported in cases of alloimmune thrombocytopenia (Kaplan, 2006; Kaplan & Freedman, 2007). We studied this protein by MD simulations (for more details, see references (Jallu et al., 2012; Jallu et al., 2013; Jallu et al., 2014)).

The β3 integrin subunit structure (Poulain & De Brevern, 2012) comes from the structure of the integrin complex (PDB 3FCS (Zhu et al., 2008)). Final structure has 690 residues and was used for MD simulations. All files mentioned below are available in the demo_paper directory from the GitHub repository (https://github.com/pierrepo/PBxplore/tree/master/demo_paper).

Protein blocks assignment

The initial file beta3.pdb contains 225 structures issued from a single 50 ns MD simulation of the β3 integrin.

PBassign -p beta3.pdb -o beta3

This instruction generates the file beta3.PB.fasta. It contains as many PB sequences as there are structures in the input beta3.pdb file.

Protein Blocks assignment is the slowest step. In this example, it took roughly 80 s on a laptop with a quad-core-1.6-GHz processor.

Protein blocks frequency

PBcount -f beta3.PB.fasta -o beta3

The above command line produces the file beta3.PB.count that contains a 2D-matrix with 16 columns (as many as different PBs) and 690 rows (one per residue) plus one supplementary column for residue number and one supplementary row for PBs labels.

Statistical analysis

Distribution of PBs

PBstat -f beta3.PB.count -o beta3 --map

Figure 3 shows the distribution of PBs for the β3 integrin. The color scale ranges from blue (the PB is not found at this position) to red (the PB is always found at this position). The β3 protein counts 690 residues. This leads to a cluttered figure and prevents getting any details on a specific residue (Fig. 3A). However, it exhibits some interesting patterns colored in red that correspond to series of neighboring residues exhibiting a fixed PB during the entire MD simulation. See for instance patterns associated to PBs d and m that reveal β-sheets and α-helices secondary structures (De Brevern, 2005).

Figure 3 Distribution of PBs for the β3 integrin along the protein sequence.

On the x-axis are found the 690 position residues and on the y-axis the 16 consecutive PBs from a to p (the two first and two last positions associated to “Z” have no assignment): (A) for the entire protein; (B) for the PSI domain only (residues 1 to 56). The dashed zone pinpoints residue 33 to 35.

With a large protein such as this one, it is better to look at limited segments. A focus on the PSI domain (residue 1 to 56) (Jallu et al., 2012; Zhu et al., 2008) of the β3 integrin was achieved with the command:

PBstat -f beta3.PB.count -o beta3 --map --residue-min 1 --residue-max 56

Figure 3B shows the PSI domain dynamics in terms of PBs. Interestingly, residue 33 is the site of the human platelet antigen (HPA)-1 alloimmune system. It is the first cause of alloimmune thrombocytopenia in Caucasian populations and a risk factor for thrombosis (Kaplan, 2006; Kaplan & Freedman, 2007). In Fig. 3B, this residue occupies a stable conformation with PB h. Residues 33 to 35 define a stable core composed of PBs h-i-a. This core is found in all of the 255 conformations extracted from the MD simulation and then is considered as highly rigid. On the opposite, residue 52 is flexible as it is found associated to PBs i, j, k and l corresponding to coil and α-helix conformations.

Equivalent number of PBs

The Neq is a statistical measurement similar to entropy and is related to the flexibility of a given residue. The higher is the value, the more flexible is the backbone. The Neq for the PSI domain (residue 1 to 56) was obtained from the command line:

PBstat -f beta3.PB.count -o beta3 --neq --residue-min 1 --residue-max 56

The output file beta3.PB.Neq.1-56 contains two columns, corresponding to the residue numbers and the Neq values. Figure 4A represents the Neq along with the PBs sequence of the PSI domain, as generated by PBstat. The rigid region 33–35 and the flexible residue 52 are easily spotted, with low Neq values for the former and a high Neq value for the latter.

Figure 4 (A) Neq versus residue number for the PSI domain (residues 1 to 56); (B) comparison between RMSF and Neq.

An interesting point, seen in our previous studies, is that the region delimited by residues 33 to 35 was shown to be highly mobile by the RMSF analysis we performed in Jallu et al. (2012). RMSF was calculated on C-α atoms on the whole protein, for more details, see ‘Materials and Methods’ section in Jallu et al. (2012). For comparison, RMSF and Neq are represented on the same graph on Fig. 4B. This high mobility was correlated with the location of this region in a loop, which globally moved a lot in our MD simulations. Here, we observe that the region 33–35 is rigid. The high values of RMSF we observed in our previous work were due to flexible residues in the vicinity of the region 33–35, probably acting as hinges (residues 32 and 36–37). Those hinges, due to their flexibility, induced the mobility of the whole loop: the region 33–35 fluctuated but did not deform. Understanding the flexibility of residues 33 to 35 is important since this region defines the HPA-1 alloantigenic system involved in severe cases of alloimmune thrombocytopenia. PBxplore allows discriminating between flexible and rigid residues. The Neq is a metric of deformability and flexibility whereas RMSF quantifies mobility.

Logo representation of PBs frequency

While the Neq analysis focuses on the flexibility of amino acids, the WebLogo-like representation (Crooks et al., 2004) aims at identifying the diversity of PBs and their frequencies at a given position in the protein sequence. With a focus on the PSI domain, the following command line was used:

PBstat -f beta3.PB.count -o beta3 --logo --residue-min 1 --residue-max 56

Figure 5 represents PBs found at a given position. The rigid region 33–35 is composed of a succession of PBs h-i-a while the flexible residue 52 is associated to PBs i, j, k and l. This third representation summarized pertinent information, as shown in Jallu et al. (2013).

Figure 5 WebLogo-like representation of PBs for the PSI domain of the β3 integrin.

PBs in red roughly correspond to α-helices, PBs in blue to β-sheets and PBs in green to coil.

Conclusion

From our previous works (Jallu et al., 2012; Jallu et al., 2013; Jallu et al., 2014; Chevrier et al., 2013), we have seen the usefulness of a tool dedicated to the analysis of local protein structures and deformability with PBs. We also showed the relevance of studying molecular deformability in the scope of structures issued from MD simulations. In a very recent study (Goguet et al., 2017), long independent MD simulations were performed for seven variants and one reference structure of the Calf-1 domain of the αIIb human integrin. Simulations were analyzed with PBxplore. Common and flexible regions as well as deformable zones were observed in all the structures. The highest B-factor region of Calf-1, usually considered as most flexible, is in fact a rather rigid region encompassed into two deformable zones. Each mutated structure barely showed any modifications at the mutation sites while distant conformational changes were detected by PBxplore. These results highlight the relevance of MD simulations in the study of both short and long range effects on protein structures, and demonstrate how PBs can bring insight from such simulations. In this context, we propose PBxplore, freely available at https://github.com/pierrepo/PBxplore. It is written in a modular fashion that allows embedding in any PBs related Python application.

Software Availability

PBxplore is released under the open-source MIT license (https://opensource.org/licenses/MIT). Its source code can be freely downloaded from the GitHub repository of the project: https://github.com/pierrepo/PBxplore. In addition, the present version of PBxplore (1.3.8) is also archived in the digital repository Zenodo (Barnoud et al., 2017).

Additional Information and Declarations

Competing Interests

Author Contributions

Data Availability

The authors declare there are no competing interests.

Jonathan Barnoud and Hubert Santuz performed the experiments, analyzed the data, contributed reagents/materials/analysis tools, wrote the paper, prepared figures and/or tables, reviewed drafts of the paper.

Pierrick Craveur, Agnel Praveen Joseph and Vincent Jallu contributed reagents/materials/analysis tools, reviewed drafts of the paper.

Alexandre G. de Brevern conceived and designed the experiments, analyzed the data, contributed reagents/materials/analysis tools, wrote the paper, prepared figures and/or tables, reviewed drafts of the paper.

Pierre Poulain conceived and designed the experiments, performed the experiments, analyzed the data, contributed reagents/materials/analysis tools, wrote the paper, prepared figures and/or tables, reviewed drafts of the paper.

The following information was supplied regarding data availability:

GitHub: https://github.com/pierrepo/PBxplore

Zenodo: https://dx.doi.org/10.5281/zenodo.1016257.

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
