# Peer review of "PBxplore: a tool to analyze local protein structure and deformability with Protein Blocks"

_PeerJ, doi:10.7717/peerj.4013_

## Round 0.1 · original submission · Minor Revisions

Both reviewers recommend minor revisions. Please see the comments below.

Reviewer 1 ·

Basic reporting

no comment

Experimental design

no comment

Validity of the findings

Some conclusion and discussion need more clarification (see below in general comments).

Additional comments

The manuscript entitled ‘PBxplore: a tool to analyze local protein structure and deformability with Protein Blocks’ by Barnoud et al presents the implementation and development of a python tool based on Protein Blocks definition to characterize local structures in a protein. The protein blocks (PBs) used in this work have been previously published and have been shown to be useful in many applications such as peptide design, local protein conformation predictions etc. The suit of tools presented by the authors can be used either for a single structure or for a trajectory. The work presented is technically sound, however a few points could be clarified when comparing the results of PBxplore and RMSF calculations for instance.
Even though more details about the RMSF calculation are present in the cited paper (Jallu et al 2012), the authors could specify that it was calculated on Calpha atoms. Since RMSF values are obtained when aligning each frame of the simulation on a selection of atoms, what would be the values if the authors align their trajectory on the loop 33 to 35? It should be clarified that indeed RMSF values if performed on all the structure including the loops for instance will give a more global view of flexibility, but if calculated with respect to a subset of atoms of interest it could give a more local picture. It is not very clear what the authors imply when stating the residues 32 and 36-37 around the loop of interest are acting as hinges. From the Neq plot it seems that indeed residue 32 has an elevated value, but it is not as striking for residues 36 and 37. Also, the sentence on page 6 ‘These results question the relationship between MD simulations and allostery and the role of long range effects on protein structure’ is a bit of an over-statement in this context, as long range effects have not been explored thoroughly in this work. It is rather the difficulty to extract from MD simulations meaningful information on structural changes which play a role in allosteric signaling.
Notwithstanding the above, it is interesting to see that regions displaying elevated global RMSF values can easily be pinpointed in a straightforward fashion using PBxplore for further analysis regarding local vs global flexibility. Therefore this tool should be a valuable tool for analysis in computational structural biology.

·

Basic reporting

no comment

Experimental design

no comment

Validity of the findings

no comment

Additional comments

It will be good to develop a statistical measure to identify the residues which are not deformable and flexible, yet their RMSFs are high (example of beta-3 integrin). It will help PB analysis on the top of RMSF. On the other hand it will also help to highlight regions in an analysis, if any, for which the RMSFs are low yet deformabilty and flexibilty are high (as around residue 20).
It will also help users if a web based server could be setup for PBxplore (should not be difficult).
Small suggestion: Name of the tool should be "PBxplorer" and not "PBxplore".

---

## Round 0.2 · accepted · Accept

The present paper describes the development and application of PBxplore to analyze dynamics of protein structures. The computational tool has focus on the analysis of dynamics and deformability of protein structures using Protein Blocks. In my view, the manuscript is interesting and reports a new computational tool to be used for those interested in molecular dynamics simulations of proteins. The authors answered all points raised by the reviewers. It can be accepted as it is.